## REVIEW ARTICLE

# Mapping lung squamous cell carcinoma pathogenesis through in vitro and in vivo models

Sandra Gómez-López [1], Zoe E. Whiteman [1,2] & Sam M. Janes [1✉]

Lung cancer is the main cause of cancer death worldwide, with lung squamous cell carcinoma (LUSC) being the second most frequent subtype. Preclinical LUSC models recapitulating human disease pathogenesis are key for the development of early intervention approaches and improved therapies. Here, we review advances and challenges in the generation of LUSC models, from 2D and 3D cultures, to murine models. We discuss how molecular profiling of premalignant lesions and invasive LUSC has contributed to the refinement of in vitro and in vivo models, and in turn, how these systems have increased our understanding of LUSC biology and therapeutic vulnerabilities.

A pproximately 2.2 million lung cancer cases are diagnosed each year (https://gco.iarc.fr/today). As most patients present with a late-stage incurable disease, lung cancer is the major contributor to cancer mortality worldwide, accounting for 18.4% of deaths[1]. The design of more effective therapeutic strategies and the development of early detection and intervention approaches are therefore global health priorities.

Around 85% of lung cancer cases correspond to the non-small-cell lung cancer (NSCLC) subtype. Among these, one-third are lung squamous cell carcinoma (LUSC), and the rest are predominantly adenocarcinoma (LUAD)[2]. Established cancer cell lines have been instrumental in the investigation of the molecular mechanisms driving NSCLC and in drug discovery studies. However, in the absence of molecular fingerprinting and routine surveillance, intra- or inter-species contamination and cell line misidentification can severely compromise the potential of these models[3]. Currently, the availability of well characterised NSCLC cell lines is biased towards LUAD[4].

Over the last decade, sequencing studies have not only identified recurrent genomic alterations across large cohorts of LUSC samples, but also highlighted the great inter- and intra-tumour heterogeneity and their complex evolutionary histories[5,6], which cannot be recapitulated in traditional cancer cell line cultures. Additionally, it is clear that the tumour microenvironment plays an essential role in lung cancer progression[7]. Together, these observations stress the importance of developing alternative LUSC models that allow dissection of the molecular pathogenesis of LUSC, analyses of complex cell-cell and cell-microenvironment interactions, and assessment of tailored therapies at different stages of disease progression.

Here, we review advances in the development of in vitro and in vivo LUSC models and their implications in understanding the cellular and molecular processes driving LUSC development and responses to therapy.

### In vitro systems

*Modelling pre-invasive disease in air—liquid interface (ALI) culture.* LUSC develops in a step-wise manner from increasingly disordered pre-invasive lesions in the bronchial epithelium. Pre-invasive lesions are associated with cigarette smoking, the main lung cancer risk factor[8–10]. The histology ranges from hyperplasia, metaplasia and dysplasia to carcinoma-in situ (CIS). Molecular studies have identified recurrent genomic, transcriptomic and epigenetic alterations in

[1] Lungs for Living Research Centre, UCL Respiratory, University College London, London, UK. [2] Cancer Research UK UCL Centre, UCL Cancer Institute, University College London, London, UK. ✉email: s.janes@ucl.ac.uk

pre-invasive squamous lung lesions[11–14]. Both amplification of the distal region of chromosome 3q and mutations in *TP53* are among the earliest changes commonly detected in pre-invasive lesions[11–15].

A number of genes located within the 3q amplicon have been highlighted as potential LUSC drivers, including SRY-box 2 (*SOX2*), protein kinase C iota (*PRKCI*), epithelial cell transforming 2 (*ECT2*), phosphatidylinositol-4,5-bisphosphate 3-kinase catalytic subunit alpha (*PIK3CA*)[15,16] and TRAF2 and NCK interacting kinase (*TNIK*)[17]. Gain- and loss-of-function studies in NSCLC cell lines have shown the transcription factor SOX2 to be a master regulator of squamous identity and a lineage-survival oncogene[18]. More recently, in vitro models have been used to investigate the effects of early *SOX2* amplification and its potential interactions with other alterations occurring during the progression of pre-invasive lesions to invasive LUSC.

Since its development[19], the ALI culture system has been widely used for functional in vitro analyses of the bronchial epithelium. Epithelial cells are seeded onto a biphasic cultivation chamber with a permeable support layered with an extracellular matrix gel substrate, with or without embedded stromal cells. Following initial growth in submerged culture, the apical surface of the epithelial cells is exposed to the air, while they keep receiving nutrients basally. After continuous culture in ALI conditions, ciliated and mucinous cell differentiation occurs, resulting in the formation of a polarised epithelial sheet (Fig. 1).

Enforced expression of *SOX2* in human bronchial epithelial cells in ALI culture has been shown to induce the formation of squamous metaplasia[20] and dysplasia[21], enhance mucinous differentiation and reduce ciliated differentiation[20]. Silencing *TP53* led to the formation of focal outgrowths within the epithelial cell sheet; when combined with *SOX2* overexpression, the outgrowths were more diffuse[21]. This indicates that early dysregulation of *SOX2* and *TP53* in the bronchial epithelium jointly drives a dysplastic phenotype. The ability of SOX2 to induce squamous metaplasia and dysplasia was found to require phosphatidylinositol 3-kinase (PI3K) activity, as either pharmacological inhibition of the PI3K/AKT pathway[20,21] or knockdown of *PIK3CA*[20]—encoding the p110α catalytic subunit of PI3K—suppressed squamous differentiation. The cooperative roles between *SOX2* and *PIK3CA* provide mechanistic insights about the selective advantage of 3q amplification during pre-invasive LUSC evolution.

*Human primary LUSC in 2D cell culture*. Primary human epithelial cells from diverse tissues can be robustly expanded in vitro when cultured on a feeder cell layer of mitotically inactivated murine 3T3 fibroblasts and in the presence of Rho kinase (ROCK) inhibitor Y-27632[22]. Airway epithelial cells grown under these conditions, frequently referred to as 'conditional reprogramming' (CR), retain features of airway basal stem cells, including multi-lineage differentiation potential[23]. These methods have been applied in the establishment of primary NSCLC cell cultures[24,25]. However, reported success rates have been variable, with various studies indicating that CR conditions preferably support the growth of normal airway epithelial cells[26–28].

In CR cultures established from resected primary NSCLC tumours, including several cases of LUSC, cancer cells were found to be outgrown by normal airway epithelial cells[26,27]. This resulted in the loss of both patient-specific cancer-associated genetic alterations[26,27] and in vivo tumourigenic potential[26]. When cultures were initiated from lung cancer brain metastases or NSCLC patient-derived xenografts, both expected to lack contaminating human epithelial cells, they failed to expand[26]. The latter suggests that CR conditions cannot sustain the in vitro propagation of NSCLC cells.

A variation of the CR protocol that uses human instead of murine fibroblasts as a feeder layer, has been reported to support the establishment of NSCLC cell cultures from effusions and biopsies when used during the initial plating step[24,29]. Cultures are subsequently passaged off the feeder layer for further expansion. The resulting cell lines were found to retain key mutations identified in the corresponding patient tumour[24,29]. Yet, this method has been mainly used for LUAD. It is possible that CR conditions may be less permissive for the propagation of cells carrying LUSC-associated genetic and/or epigenetic changes. Side-by-side comparison of the two feeder cell types should help clarify the applicability of CR conditions both during the derivation and long-term expansion of primary LUSC cell cultures.

*3D models*. When embedded in basement membrane matrix hydrogels (e.g., Matrigel) and cultured in non-adherent conditions, either in the presence or absence of supporting stromal cells, adult airway stem and progenitor cells can give rise to self-organising 3D hollow structures called 'organoids'. Lung organoid cultures may be established from primary murine or human cells and phenocopy basic cellular aspects of the airway epithelium, with polarised progenitor cells and their differentiated progeny arranged around a lumen[30] (Fig. 1). This system has been used for the development of in vitro LUSC models.

*Validating tumour suppressors and oncogenes in 3D culture*. In addition to mutations in the tumour suppressor *TP53* and amplification of chromosome 3q, molecular profiling studies of LUSC samples have identified significant dysregulation of a number of pathways, including cyclin-dependent kinase inhibitor 2A (*CDKN2A*)/retinoblastoma 1 (*RB1*), PI3K/AKT, squamous differentiation, and oxidative stress response[6,18]. The phenotypic consequences of several of these alterations, as well as potential synergistic effects, have been investigated using murine lung organoid models (Figs. 1 and 2).

Inactivation of *Trp53* (the mouse orthologue of *TP53*) in bulk mouse tracheal epithelial cells has been shown to enhance organoid formation efficiency[31]. When 3D cultures were initiated with purified basal cells, p53 deficiency increased both organoid number and size and enabled longer-term passaging, indicative of aberrant stem/progenitor cell self-renewal[32]. Depletion of Kelch-like ECH associated protein (KEAP1), a negative regulator of the oxidative stress response, also augmented organoid forming capacity of tracheal epithelial cells[31]. Concurrent inactivation of *Keap1* and *Trp53* resulted not only in greater increase of organoid number than individual gene deletion, but also in the formation of solid spheres, suggesting synergistic effects of the two genetic alterations. The loss of *Keap1* in *Trp53*-deficient organoids lowered intracellular reactive oxygen species (ROS) levels. This was found to mediate the increase in organoid forming ability, to enhance the metastatic potential of organoid-derived cells upon implantation in mice, and to confer resistance to oxidative stress and radiotherapy[31].

Using lentiviral vectors to model amplification of different genes located within the 3q26 chromosomal region, it has been shown that *SOX2*, *ECT2* and *PRKCI* exert cooperative interactions to induce neoplastic transformation of *Trp53*-depleted murine basal cells[32]. In organoid cultures, *SOX2*, *ECT2*, and *PRKCI* were found to each have a dominant role in the control of basal cell self-renewal, cell proliferation and epithelial polarity, respectively. Simultaneous overexpression of the three genes in p53-deficient basal cells led to increased number and size of organoids and acquisition of disorganised solid morphology. When grafted into mice, these cells gave rise to tumours with features of LUSC[32]. As both SOX2 and ECT2 are phosphorylation

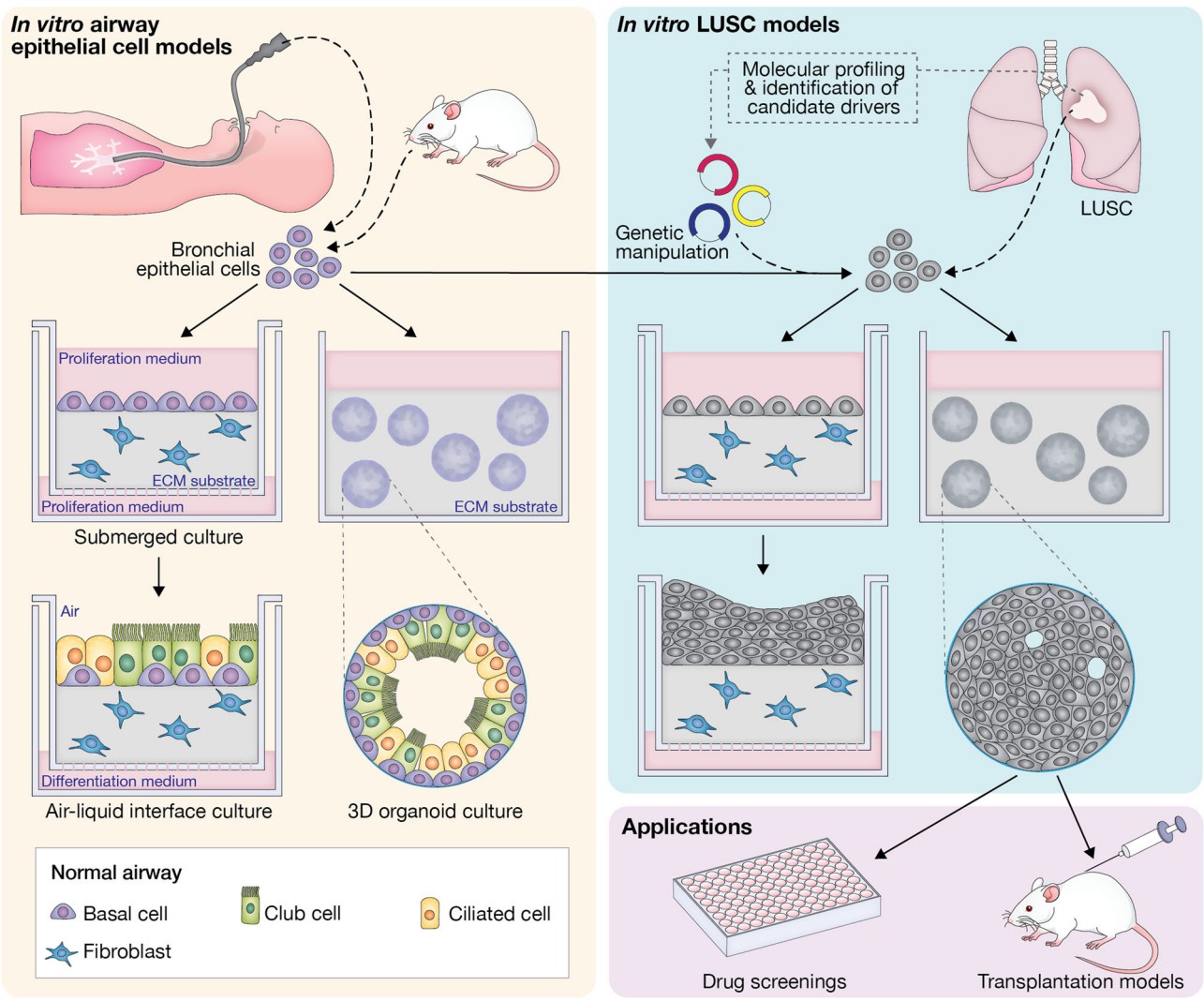

**Fig. 1 In vitro models of lung cancer and their application in in vivo studies.** Establishment of air—liquid-interface (ALI) and organoid cultures from human or mouse airway epithelial cells and LUSC tissue. Following ALI or 3D culture, normal airway epithelial basal cells produce pseudostratified epithelial sheets or hollow organoids containing differentiated cells, respectively. In contrast, LUSC cells give rise to epithelial sheets with features of dysplasia and more solid, disorganised organoids. Cultured cells may be subjected to genetic and pharmacological manipulation to investigate the phenotypic consequences of molecular alterations recurrently identified in LUSC samples. Organoids can be used in in vitro drug screenings and may be implanted into mice to evaluate their ability to give rise to tumours in vivo and response to therapies. ECM extracellular matrix.

targets of PRKCI[33,34], phosphorylation-resistant mutant forms of each gene were used in organoid cultures to demonstrate that the effects of PRKCI on airway epithelial cell polarity and proliferation are mediated by SOX2 and ECT2, respectively[32]. The latter emphasises the multifunctional role of SOX2 in early LUSC development.

In a different study, 3D cultures were established from bulk bronchial cells isolated from mice with Cre-inducible expression of *Sox2* and *Cas9*. The resulting organoids were genetically manipulated in vitro to both induce *Sox2* overexpression and have CRISPR/Cas9-mediated loss-of-function mutations in *Trp53*, *Cdkn2a,* and phosphate and tensin homologue (*Pten*), a negative regulator of PI3K signalling. Longitudinal analysis of organoid diameters showed that double- and triple-mutant organoids displayed a growth advantage. Upon engraftment into mice, triple-mutant organoids formed tumours reminiscent of well-differentiated LUSC[35].

Overall, organoid cultures, together with the diverse array of currently available genetic manipulation techniques, constitute a powerful tractable platform to dissect molecular drivers and

vulnerabilities of LUSC in vitro. Furthermore, the ability to assess organoid tumour-forming potential in vivo following transplantation into murine hosts makes murine organoid cultures a valuable tool for the generation of immunocompetent LUSC mouse models (Fig. 1), as we discuss later.

*Patient-derived lung cancer organoids.* Organoid culture methods for tissue stem cells have been adapted for the 3D expansion of patient-derived tumour cells from different epithelial cancers (Fig. 1). To prevent normal cells from outgrowing cancer cells, culture conditions need to be selective or organoids initiated exclusively with tumour cells[36]. In contrast to organoids derived from normal bronchial epithelial cells, NSCLC-derived 3D cultures produce more solid spheres lacking apico-basal polarity[37].

Different strategies have been used for the derivation of patient-derived lung cancer organoids. One approach exploited the high prevalence of *TP53* mutations in NSCLC[6,38] and used the TP53 stabilising drug Nutlin-3a to selectively inhibit the growth of cells with wild-type TP53, thereby enriching for mutant tumour cells[39]. In an alternative approach, tumour-derived

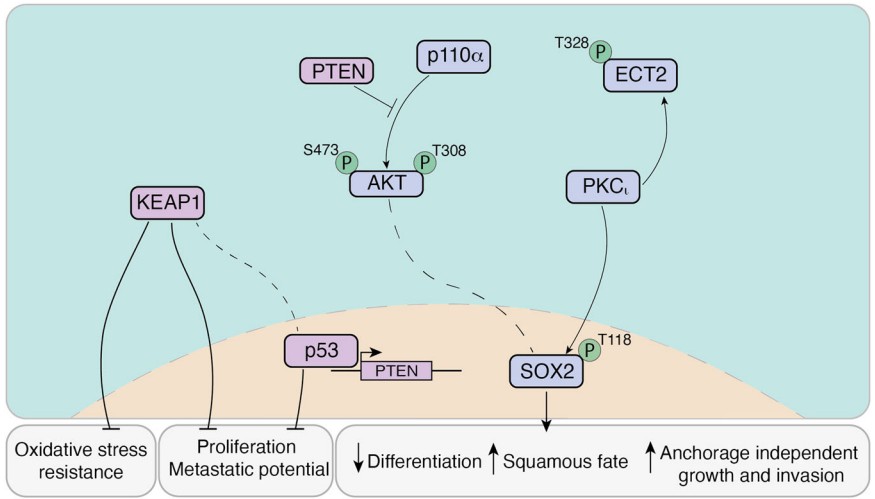

**Fig. 2 Interaction of signalling pathways demonstrated in in vivo and in vitro models of LUSC.** SOX2, ECT2, PKCι (encoded by *PRKCI*), and PI3K signalling cooperate to promote a neoplastic cell fate in LUSC models. AKT is a downstream effector of stimulated p110α. Full AKT activation is achieved when phosphorylated at both positions S473 and T308[127]. High levels of SOX2 have been correlated with upregulated phospho-AKT[21]. PKCι phosphorylates and directly interacts with ECT2 to promote anchorage-independent growth and invasion through downstream targets. PKCι phosphorylates SOX2 favouring squamous cell fate and decreased differentiation[16]. Loss of p53, PTEN, and KEAP1 have been used to model LUSC phenotypes both in vitro and in vivo. Simultaneous loss of p53 and KEAP1 has shown synergistic effects, inducing increased proliferation, metastatic potential, and resistance to oxidative stress[31]. p53 activity can inhibit PI3K signalling through PTEN-dependent and potentially-independent mechanisms in squamous cell carcinomas[128]. Additional interactions between depicted proteins have been described in other cellular contexts.

organoids were established from five different lung cancer types, including LUSC, by using a minimum basal medium (MBM) that does not support the expansion of normal airway epithelial cells[40]. To generate patient-matched normal control lung organoids, the MBM was supplemented with Wnt family member 3A (WNT3A) and inhibitors of the transforming growth factor-beta (TGFβ) and bone morphogenic protein (BMP) signalling pathways. With both methodologies, LUSC-derived organoids retained the histology and overall pattern of genomic alterations of the original tumour[39,40]. Yet, a subset of LUAD-derived organoids exhibited higher variant allele frequencies than the original tumours[40]. While this was suggested to be the result of in vitro tumour cell enrichment, rather than of the acquisition of novel mutations ex vivo, the possibility of ongoing tumour evolution in organoid cultures cannot be ruled out.

Following cryopreservation, around 75% of LUSC-derived organoids can be successfully reconstituted, enabling the generation of biobanks[40]. In drug screening trials, the response of individual lung cancer organoids varied according to their mutational profile, highlighting their potential as tools for predicting patient-specific therapeutic responses, as well as for drug development studies[39,40]. However, given the propensity of NSCLC-derived organoid cultures to be outgrown by normal cells, tumour purity should be verified regularly through a combination of strategies, including immunostaining and genetic profiling[41].

### Autochthonous murine models of LUSC

*Chemical carcinogenesis.* In vivo lung cancer modelling was at first limited to the study of spontaneously developing lung cancers in susceptible mouse strains such as A/J and SWR[42]. These strains have since been widely exploited in chemical carcinogenesis models[43]. Most chemical models have focused on the application of cigarette smoke[44,45] and its toxic components to induce LUSC[46]. Exposing female B6C3F1 mice to cigarette smoke was shown to induce broad airway carcinogenesis, predominantly adenoma, and adenocarcinoma, but very few incidences (4.2%) of

LUSC[47]. Individual toxic cigarette smoke components have shown more specific induction of LUSC development.

Benzo(a)pyrene, a polycyclic aromatic hydrocarbon commonly found in the environment, cigarette smoke, and coal tar, has shown an almost exclusive induction of LUSC in ~80% of C3H/He mice when administered intratracheally alongside charcoal powder. Yet, this selective induction of LUSC is highly dose- and regimen-dependent, with 74% of incidences characterised as either LUAD or papillary-type adenomas when the dose of Benzo(a)pyrene was halved. Doubling the dosing regimen length restored almost exclusive induction of LUSC (87% of mice) at the lower dose of Benzo(a)pyrene. Similar results were observed in the C57BL/6 strain[48]. 3-methylcholanthrene (MCA), another polycyclic aromatic hydrocarbon has been shown to induce LUSC when intratracheally injected into BC3F1 mice with 86% incidence, 2.5−6.5 months following treatment. However, induction is strain- and regimen-dependent with LUSC observed only in 6% of DBA/2 mice 7 months after treatment[49]. Ultimately, it is likely that the necessity of intratracheal administration of polycyclic hydrocarbons has favoured the use of topically applied agents such as nitrosourea derivatives to study LUSC progression.

Since the discovery of the toxicity of nitrosamine compounds, commonly found in processed meats and cigarette smoke, they have been used in models of several types of cancer[50–52]. N-nitroso-tris-chloroethylurea (NTCU), a triple chlorinated nitrosamine derivative, has an almost exclusive propensity to induce the development of LUSC when topically applied to female Swiss mice compared to other nitrosoalkylureas, which widely trigger the development of skin carcinomas, LUAD and leukaemia[50]. Despite this, dosing regimen, treatment length (2−32 weeks), mouse strain and sex all affect NTCU-induced carcinogenesis[53–55]. Overall, SWR/J, NIH Swiss, A/J, BALB/cJ, and FVB/NJ have been shown to be sensitive to NTCU, with females displaying higher susceptibility than males. Resistant strains included AKR/J, 129/svJ, and C57BL/6J[56].

The epithelial histological changes induced by NTCU application are representative of the step-wise evolution of human LUSC[52,54,57]. Early NTCU-induced changes have been shown to

begin in the trachea, with Keratin 5 (KRT5)[+] and KRT14[+] tracheal basal cell dysplasia and loss of Club and ciliated cells prior to bronchial dysplasia development[57]. NTCU-induced pre-invasive lesions express oxidative phosphorylation proteins at higher levels than control-treated lungs, akin to human bronchial pre-invasive lesions, indicating increased cellular respiration[58].

Linkage disequilibrium genetic mapping identified three genetic loci significantly associated with NTCU susceptibility: D1Mit169, D3Mit178, and D18Mit91, which are syntenic to the human chromosome loci 6q12−14, 3q26.2−26.31, and 5q23−31, respectively[56]. This suggests that NTCU-induced LUSC may resemble human LUSC genetically, as 3q26 amplification is the most frequent human LUSC genetic alteration[16] and losses in chromosome 5q have been described as common early clonal events, potentially key for LUSC induction[5]. Whole-exome sequencing of NTCU-induced lung tumours in NIH-Swiss mice revealed that 80% (47/59) of recurrently mutated genes were also altered in human LUSC tumours, these included Kmt2d, Zeb2, Braf, and Igf2r[59]. Analysis of transcriptional profiles of laser capture microdissected NTCU-induced LUSC in NIH-Swiss mice has shown that the top 150 dysregulated genes in human LUSC are also altered in NTCU-induced LUSC[53]. RNA sequencing studies of airway brushings from NTCU-treated mice have linked the PI3K pathway and activation of nuclear factor kappa B (NFκB) with NTCU-induced LUSC[60]. The transcriptomic similarity with human LUSC has allowed NTCU-induced LUSC gene expression to be included in a dataset alongside human bronchial biopsies, brushes, and TCGA LUSC tumours, to identify four molecular subtypes of human premalignant lesions[61].

Although further understanding of the genetic triggers and evolution in chemical models is still required, it is clear that NTCU-induced LUSC has the potential to recapitulate some of the heterogeneity, mutational burden, and tumour−microenvironment interactions occurring during different stages of human LUSC development, making it an ideal substrate for intervention trials. Chemopreventive studies thus far have focused on evaluating the potential of natural herbs[62], vitamin levels[63], antioxidants[64], and anti-inflammatory agents[65] to protect against NTCU-induced LUSC development.

*Genetically-engineered mouse models.* The high genomic complexity of LUSC and the extensive inter- and intra-tumour heterogeneity[5,6] have hindered the development of genetically-engineered mouse models (GEMMs) that accurately recapitulate the molecular alterations and histopathology of the human disease. Yet, continuous efforts are being made to generate clinically relevant transgenic LUSC models (Table 1).

Expression of Kirsten rat sarcoma viral oncogene (Kras) mutant $Kras^{G12D}$ in the murine airway epithelium is sufficient to drive LUAD formation[66]. However, when combined with additional somatic alterations, mice may develop tumours with features of LUSC. Deletion of the tumour suppressor serine/threonine kinase 11 (Stk11, also known as Lkb1) in the context of oncogenic KRAS leads to the formation of LUAD, lung adenosquamous tumours (LUASC), and LUSC[67]. Longitudinal analyses of this model revealed that LUSC arises from squamous differentiation of LUAD lesions, with LUASC constituting an intermediate state[68]. This transition, which is accompanied by epigenetic de-repression of Sox2, ΔNp63, nerve growth factor receptor (Ngfr), and Krt5, as well as decreased collagen deposition-associated extracellular matrix (ECM) remodelling, also occurred following Stk11 ablation in established $Kras^{G12D}$-induced LUAD[68,69]. However, mutations or downregulation of STK11 are rare in human LUSC (Fig. 3) and more prevalent in LUAD[38].

Depletion of TGFβ receptor 2 (Tgfbr2) concurrently to $Kras^{G12D}$ activation substantially increased LUSC incidence, without eliminating LUAD formation[70]. LUSC development was associated with dampened extracellular signal-regulated protein kinase 1 and 2 (ERK1/2) activity and upregulation of Sox2. While LUSC appeared predominantly along the bronchioles, LUAD tumours were located in the alveolar space, resembling human pathology. Similar spatial segregation of the two tumour types was seen when lung tumourigenesis was induced by the concomitant activation of $Kras^{G12D}$ and ablation of F-box/WD repeat-containing protein 7 (Fbxw7)[71]. In this model, LUSC and LUAD arose at 1:1 ratio, enabling side-by-side comparison of cisplatin chemotherapy response in both tumour types. There are, however, caveats to the use of $Kras^{G12D}$ in GEMMs of LUSC. Activating mutations of KRAS are around ten times more prevalent in LUAD than in LUSC[6,38]. Additionally, multi-region whole-exome sequencing analyses indicate that, when present in LUSC, these tend to occur late during tumour evolution[5].

Although rarely affected by genetic alterations, the serine/threonine kinase component of inhibitor of nuclear factor kappa B kinase complex (CHUK, also known as IKKA) is downregulated in a fraction of human LUSC cases. Knock-in mice expressing a kinase-dead form of CHUK ($Chuk^{K44A}$) develop severe skin lesions and spontaneous LUSC[72]. Re-introduction of wild-type Chuk under the control of the Loricrin promoter rescued the skin phenotype, while LUSC developed with 100% penetrance. Tumours exhibited elevated ERK activity and decreased p53 levels and their formation was preceded by increased expression of ΔNp63 and tripartite motif-containing 29 (TRIM29), inflammation, and lung enlargement.

Additional GEMMs have been developed by targeting components of core pathways significantly altered across large cohorts of LUSC patients. Alterations in the PI3K/AKT pathway are seen in around half of LUSC cases[6]. PTEN is mutated or deleted in ~21% of LUSC cases (Fig. 3), and these alterations tend to occur during the early stages of tumour evolution[5]. Ablation of both Pten and Stk11 in the airway epithelium induced LUSC formation with high penetrance, although some phenotypic variations were seen depending on the induction method[73,74]. Pten- and Stk11-deficient LUSC formed both near the main bronchi and at peripheral locations and exhibited high levels of phospho-AKT and low ERK activity[73,74]. A subpopulation of tumour cells, identified by co-expression of epithelial cell adhesion molecule (EPCAM), the basal cell marker NGFR, and the bronchioalveolar stem cell (BASC) marker stem cell antigen-1 (SCA1, also known as LY6A), was found to have the ability to propagate the tumour upon implantation into immunocompromised mice[73]. However, Sca1 does not have a human orthologue, so the relevance of this cellular population for human cancer has not to be established.

SOX2 gains and upregulation occur in ~40% and ~77% of LUSC cases, respectively (Fig. 3). Enforced expression of Sox2 in the murine airway epithelium, however, is not enough to drive LUSC formation[75,76]. Instead, it can result in bronchial hyperplasia and eventually LUAD with aberrant TRP63 expression[75]. Yet, a few genetic alterations have been found to cooperate with Sox2 to promote LUSC formation. Overexpression of Sox2 coupled with the loss of Stk11 induces predominantly the formation of peripheral LUSC, with a fraction of tumours expressing low levels of NK2 homeobox 1 (NKX2-1)[77,78]. Further co-deletion of Nkx2-1 in this model, significantly reduced tumour latency[78]. Longitudinal analyses suggested that the combination of enforced Sox2 expression with co-ablation of Stk11 and Nkx2-1 caused the formation of early mucinous LUAD lesions that differentiated into LUSC over time[78]. This is reminiscent of what

Table 1 Genetically-engineered mouse models of LUSC.

| Genetic alteration (Reference) | Method of induction | LUSC incidence (% of total mice) | Latency (months) | Metastasis (n) | Immune microenvironment |
|---|---|---|---|---|---|
| Chuk^K44A/K44A, Tg(Loricrin-CHUK) [72] | Not inducible | 100% | 4–6 | None detected (Chuk^K44A/K44A mice lack lymph nodes) | Enrichment of CD4+ T cells and F4/80+ macrophages and moderate increase of CD8+ T cells and Ly6G+CD11b+ neutrophils in mutant lungs; upregulation of pro-inflammatory cytokine genes |
| Kras^LSL-G12D, Stk11^f/f and Kras^LSL-G12D, Stk11^f/− [67,68] | Intranasal instillation of Ad-Cre | ~56% (includes LUASC) | 2–3 | Lymph node (27/44) and axial skeleton (4/44), all with features of LUAD | n/s |
| Kras^FSF-G12D, Stk11^f/f, Rosa26^FSF-CreERT2 [69] | Intranasal Ad-Flpo and i.p. tamoxifen | ~40% (LUAD present in 100%) | 4.5–5.5 | Lymph node (~35%) | MPO+ TANs |
| Kras^LSL-G12D, Fbxw7f/f [71] | Intratracheal Ad5-CMV-Cre or Ad5-Scgb1a1-Cre | 100% (LUAD present in alveolar space) | 2–3 | n/s | Ly6G+ TANs; PDL1 and PD1 expression in tumours |
| Kras^LSL-G12D, Tgfbr2f/f [70] | Intranasal instillation of Ad-CMV-Cre | ~95% (LUAD present in alveolar space) | 1.5–2 | Mediastinal lymph node, heart, and kidneys (14/21 at 4.5 months), all with features of LUSC | Infiltrating MPO+ and Ly6G+ TANs; upregulation of neutrophil chemoattractant genes (CXC ligands) |
| Stk11f/f and exogenous Sox2 [77] | Intranasal instillation of bicistronic lentivirus expressing Actb-Sox2 and Pgk-Cre | ~35% (based on biomarker staining) | 6–10 | n/s | n/s |
| Stk11f/f; R26^LSL-Sox2-IRES-GFP [78] | Intranasal instillation of Ad5-CMV-Cre | ~71% | 11 | n/s | Infiltrating CD11B+, MPO+ and Ly6G+ TANs; FOXP3+ Tregs in tumours; upregulation immunosuppressive genes (Vtcn1, Cd80, Btla, Havcr2, and Pdl1), and downregulation of antigen presentation genes in tumours |
| Stk11f/f; R26^LSL-Sox2-IRES-GFP, Nkx2-1f/f [78] | Intranasal instillation of Ad5-CMV-Cre | 100% (LUAD present at lower prevalence) | 2–4 | n/s | Infiltrating CD11B+, MPO+ and Ly6G+ TANs; CXCL5 expression in tumours |
| Scgb1a1Cre, Stk11f/f; Mapk8f/f; Mapk9−/− [74] | Constitutive Cre expression from the Scgb1a1 locus (Cre expression is detected in a subpopulation of TRP63+ tracheal basal cells in this Cre driver line) | ~33% | 11 | n/s | n/s |

## Table 1 (continued)

| Genetic alteration (Reference) | Method of induction | LUSC incidence (% of total mice) | Latency (months) | Metastasis (n) | Immune microenvironment |
|---|---|---|---|---|---|
| *Stk11^f/f; Pten^f/f* ([73]) | Intranasal instillation of Ad-Cre | 100% | 9–11.5 | Chest wall (3/78) | Infiltrating Ly6G⁺ and MPO⁺ TANs; FOXP3⁺ Tregs, CD4⁺ and CD8⁺ T cells in tumours and surroundings; PD1 expression in subpopulations of CD4⁺ and CD8⁺ cells in tumours and stroma; PD1 expression in tumours; elevated levels of CXC ligands, TGFβ and IL6 in BAL |
| *Pten^f/f; Cdkn2ab^f/f; Col1a1^LSL-Sox2* ([79]) | Intratracheal Ad5-Krt5-Cre, Ad5-Krt14-Cre, Ad5-Sftpc-Cre, or Ad5-Scgb1a1-Cre, following naphthalene treatment | ~73% (Krt5-Cre & Krt14-Cre); 100% (Sftpc-Cre & Scgb1a1-Cre) | 7–9 | Heart (1/6) | Infiltrating Ly6G⁺ and MPO⁺ TANs; CD4⁺ and CD8⁺ T cells in tumours and stroma; PD1⁺ immune cells in stroma; PDL1 expression in tumours |

*Abbreviations. BAL bronchoalveolar lavage, f floxed, FSF FRT-STOP-FRT, i.p. intraperitoneal, LSL loxP-STOP-loxP, n/s not stated, TAN tumour-associated neutrophil.*

occurs in the *Kras^G12D*; *Stk11^−/−* model[68], and supports a previously identified role for NKX2-1 in suppressing squamous differentiation[76]. Indeed, ablation of *Nkx2-1* with concurrent *Sox2* overexpression has been shown to be sufficient to drive LUSC formation in mice[76].

The great majority of LUSC cases display cell-cycle dysregulation through alterations in the tumour suppressors genes *CDKN2A* and *RB1*[6]. Deletions or mutations in *CDKN2A* and/or its neighbouring gene *CDKN2B* occur in approximately 42% of LUSC tumours (Fig. 3). Concurrent ablation of *Cdkn2ab* and *Pten* in the murine airway induced the formation of a range of lung neoplasias, but no instances of LUSC were observed[79]. In contrast, co-deletion of *Cdkn2ab* and *Pten* concomitant with enforced expression of *Sox2* resulted solely in pre-invasive squamous lesions and LUSC[79], highlighting the key role of SOX2 in promoting a squamous fate. In an alternative model where PI3K dysregulation is driven by expression of a constitutively active form of p110α (*Pik3ca^H1047R*), concurrent overexpression of *Sox2* and loss of *Cdkn2ab* also led to the formation of LUSC with high penetrance[80]. Ablation of nuclear receptor binding SET domain protein 3 (*NSD3*), frequently upregulated in human LUSC, significantly extended animal survival and reduced tumour size in this model, indicating an oncogenic role for this methyltransferase in LUSC[80].

## Transplantation models

*Allogenic and syngeneic models.* Murine airway epithelial cells may be genetically modified in vitro and transplanted either as allografts into immunocompromised mice or into syngeneic immunocompetent mouse hosts to evaluate their ability to form tumours in vivo. Immunocompromised host strains used in transplantation models include athymic nude (nu/nu), non-obese diabetic/severe combined immunodeficiency (NOD/*scid*), and NOD/*scid*/*Il2rg*^null (NSG). These systems have been used as an alternative to GEMMs for the phenotypic characterisation of candidate LUSC driver genes, assessment of genetic interactions, investigation of tumour cell-of-origin, and evaluation of therapeutic candidates (Table 2). Syngeneic models have also been established using tumours or cell lines derived from chemically-induced or transgenic murine LUSC models[32,35,81–83] (Table 2). The derivation of cell lines from murine LUSC normally involves serial passaging in vivo and in vitro to select for cell populations capable of engraftment.

Before implantation, grafts can be modified to express a luciferase reporter, allowing for non-invasive monitoring of tumour growth and metastasis via bioluminescence imaging[32,81,84,85]. Implantation site may vary, but common establishment routes include subcutaneous graft, intravenous inoculation, and orthotopic injection in the lungs. LUSC allogenic and syngeneic models have been successfully used for the evaluation of therapeutic response to conventional, targeted, and combination therapies[31,35,85,86] (Table 3).

*Xenograft models.* Xenograft models involve the injection of either human cancer cell lines (CDXs) or patient-derived tissue (PDXs) into immunocompromised mice. After implantation, often in combination with a basement membrane mix, cells are left for tumour engraftment and growth. Xenografts have been applied across many types of cancer[87,88]. Serial passaging of orthotopic CDXs derived from the LUSC cell lines H520 and SKMES-1 has produced subclones that metastasise to LUSC characteristic sites (i.e., adrenal glands and chest wall)[89] and could be used as a xenograft models of LUSC metastasis. However, CDXs are likely to be less reflective of human lung cancers due to inherent in vitro biases on top of xenograft selection.

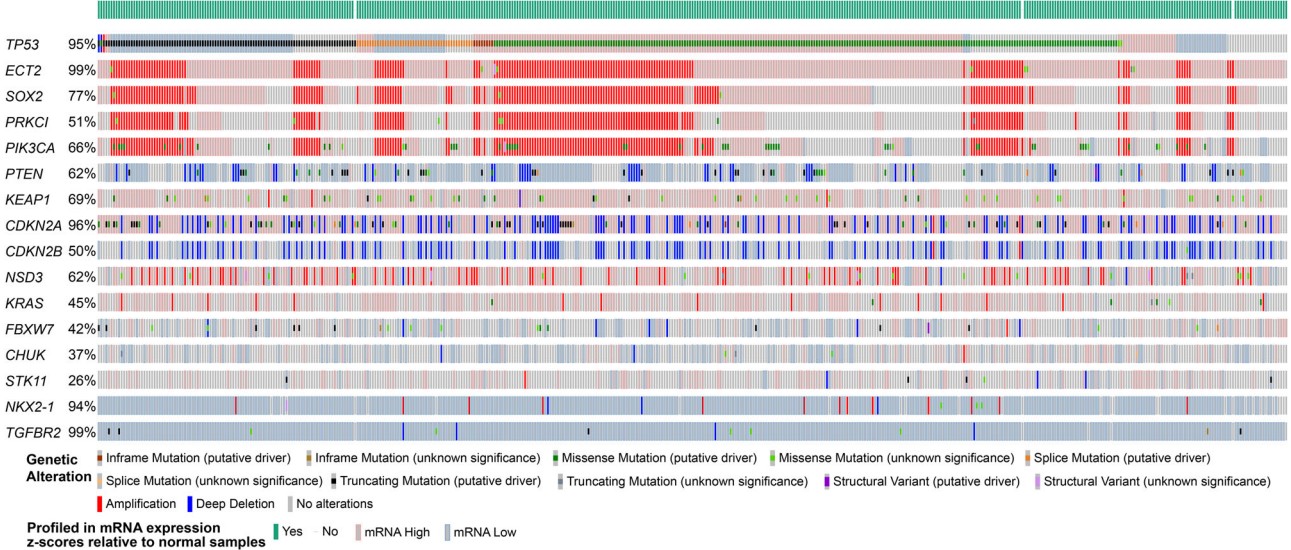

**Fig. 3 Genetic alterations used in the generation of in vitro and in vivo models of LUSC.** Oncoprint showing frequency of genetic and transcriptional changes in the indicated genes across 469 lung squamous cell carcinoma samples from human donors included in The Cancer Genome Atlas (TCGA) PanCancer Atlas dataset. Normal adjacent tissue samples in the cohort were used as a reference for gene expression changes (z-score threshold ± 2.0) (downloaded from https://www.cbioportal.org/).

To increase engraftment rates of injected cells or tumours, host-cell rejections are minimised by using immune-deficient mouse strains. NSG mice are reported to have higher engraftment rates, across cell types, than NOD/*scid* mice. NOD/*scid* mice have a complete loss of both T and B cells and impaired functioning of NK cells, whereas NSG mice are more immune-deficient with additional impaired cytokine signalling, requiring costly higher biosecurity measures (www.jax.org). In an effort to reduce the cost and time spent generating PDXs for preclinical cancer modelling the EurOPDX consortium initiative is establishing a large collection of PDX models across 30 pathologies, including NSCLC (www.europdx.eu).

Subcutaneous implantation rates for LUSC PDXs have been recorded as high as ~45% in NOD/*scid* mice[90], whereas renal capsule implantation of patient NSCLC tumours have shown engraftment rates of up to 90% in the same murine strain[91]. Engraftment rates are dependent on primary tumour and microenvironment characteristics, with poorly differentiated, larger, higher-grade tumours more likely to engraft[92]. Choice of xenograft implantation site is often a trade-off between tumour engraftment rate and ease of tumour development tracking. Subcutaneous implantation is most common due to the ease of tumour growth tracking. However, this limits the recapitulation of normal tumour−microenvironment interaction and hence is unlikely to directly follow normal LUSC development. Orthotopic xenograft models offer the potential of modelling the tumour microenvironment[93]. Yet, orthotopic endobronchial implantation of lung cancer cell lines has been shown to have much higher tumour-related mortality rates than subcutaneous implantation[94].

NSCLC PDXs have been shown to be representative of the original patient tumour histologically over serial passaging[95]. Patient tumour copy number alterations are well conserved throughout PDX engraftment and passaging. With copy number variations likely to originate from patient intratumoural heterogeneity[96]. Ninety-three percent of mutations found in the original tumours were maintained in NSCLC PDXs. However, allele frequencies varied (>1.5 fold), and PDXs harboured a significant amount of additional mutations not found in the original patient tumour with differences attributed to either tumour heterogeneity or PDX-induced tumour evolution[90].

Transcriptomic comparison between NSCLC patient tumours and tumour-derived PDXs have demonstrated a high degree of correlation[97,98]. The immune compartment represents a substantial difference between PDXs and their corresponding patient tumours, with immune-response and extracellular matrix gene signatures being the top differentially expressed[97,99]. The ability of PDXs to broadly recapitulate both the genome and transcriptome of original patient tumours presents an opportunity for individualised therapy screening. This has motivated the use of PDXs for the derivation of tumour cell lines and long-term organoid cultures from patient samples[100,101].

NSCLC PDXs have been found to be good predictors of therapeutic outcome and metastases, with 6/7 patients that experienced recurrence or metastases shown to have PDXs non-responsive to corresponding therapies[91]. LUSC PDXs have been used to improve the stratification of patients likely to respond to fibroblast growth factor receptor (FGFR) inhibitors. Contrary to previous clinical trials that used FGFR1 amplification, a genetic change that occurs in ~17% of human LUSC (cBioPortal, TCGA PanCancer Atlas), as a predictor of inhibitor response, PDX responses demonstrated that high FGFR1 RNA levels instead were indicative of FGFR inhibitor-induced tumour cell differentiation and enhanced tumour cell death, particularly when combined with cisplatin[99].

Comparative studies of LUSC PDXs carrying different PI3K pathway alterations indicate that PDXs expressing mutant *PIK3CA* are more likely to respond to the PI3K inhibitor BKM120 than those with *PIK3CA* amplification or *PTEN* loss. BKM120-responsive mutant *PIK3CA* PDXs were further found to display loss of *CDKN2A*, which encodes the cell-cycle regulator p16, an inhibitor of cyclin-dependent kinases 4 and 6 (CDK4/6) activity. When compared to single therapies, combined PI3K and CDK4/6 inhibition, resulted in enhanced anti-tumour effects, including tumour regression[102].

**LUSC cell-of-origin.** The resemblance of LUSC to airway basal epithelial cells highlighted them as the suspected cellular origin of this tumour type. However, this view has been challenged by studies in murine LUSC models.

Table 2 Allogenic and syngeneic murine LUSC models.

| Genetic alteration (reference) | Method of modification | Graft | Host | Site | Time for growth |
|---|---|---|---|---|---|
| Trp53fl/fl; Keap1fl/fl; R26tdTomato or Krt5CreERT2; Trp53fl/fl; Keap1fl/fl; R26tdTomato [31] | In vitro transduction with Ad-Cre or in vivo recombination following i.p. tamoxifen | Bulk tracheal epithelial cells or purified basal cells | Allogenic, immunocompromised (NSG) | Subcutaneous | 2–4 months |
| R26-SL-Sox2-IRES-GFP, lgs2LSL-Cas9 and knockout of Trp53, Pten, and Cdkn2a [35] | In vitro transduction with lentivirus expressing Cre and sgRNAs targeting Trp53, Pten and Cdkn2a | Tracheal cells expanded in 3D culture | Allogenic, immunocompromised (nu/nu) | Subcutaneous | 2–3 months |
| | | JH716 cell line derived from primary subcutaneous tumour | Syngeneic (C57BL/6) | Subcutaneous or orthotopic | n/s |
| Trp53fl/fl; R26LSL-Luc and exogenous SOX2, PRKCI, and ECT2 [32] | In vitro transduction with Ad-Cre and lentivirus expressing SOX2, PRKCI and ECT2 | Basal cells expanded in 3D culture | Syngeneic (C57BL/6) | Orthotopic | 2 months |
| ChukK44A/K44A, Tg (Loricrin-CHUK) [81] | GEMM (not inducible) | KAL-LN2E1 cells, a metastatic sub-clone of the KALLU cell line[72] derived after two rounds of in vivo selection | Syngeneic (FVB) | Orthotopic | n/s |
| n/d [82] | MCA-induced carcinogenesis | KLN205 cell line obtained following in vivo and in vitro passaging of primary carcinoma[49] | Syngeneic (DBA/2) | Subcutaneous (or intravenous) | n/s (~3 weeks for lung nodules) |
| n/d [83] | NTCU-induced carcinogenesis | UN-SCC680AJ cell line derived after serial in vitro and in vivo passaging of primary LUSC | Syngeneic (A/J) | Subcutaneous | 3–6 weeks |

Abbreviations. LSL loxP-STOP-loxP, n/s not stated, n/d not defined.

In the great majority of inducible GEMMs of LUSC, driver mutations are activated via intranasal or intratracheal delivery of virus expressing Cre under the control of a constitutive promoter (e.g., cytomegalovirus, CMV) (Table 1). When used on the naïve airways, these methods preferentially lead to infection of secretoglobin family 1A member 1 (SCGB1A1)$^+$ Club cells in the distal airways[31,70,79], introducing bias for cell-of-origin investigation. The use of transgenic murine lines driving basal cell-specific tamoxifen-inducible Cre expression is frequently unfeasible for lung carcinogenesis studies, as it may result in faster cancer formation in other epithelia[31,71]. In autochthonous LUSC models, these limitations have been overcome by using cell-type-restricted Cre-expressing viruses and/or by depleting the airways from luminal cells (using naphthalene or SO$_2$) to allow infection of the underlying basal cells[31,71,79]. This has allowed assessment of the ability of distinct airway epithelial subpopulations to initiate LUSC in response to defined genetic drivers.

Lineage tracing analyses following Ad-CMV-Cre administration have tracked the origin of murine LUSC tumours induced by activation of $Kras^{G12D}$ and $Tgfbr2$ ablation to SCGB1A1$^+$ Club cells[70]. Targeting the same hits to $Krt5$-expressing basal cells resulted in substantially lower LUSC incidence[103]. Similarly, when cell-type-restricted Cre expressing viruses were used to simultaneously activate $Kras^{G12D}$ and delete $Fbxw7$ in distinct subpopulations of airway epithelial cells, LUSC only developed when targeting SCGB1A1$^+$ Club cells. KRT5$^+$ basal cells did not produce any tumours, whereas ciliated (Forkhead box J1, FOXJ1$^+$) and alveolar Type II (Surfactant protein C, SFTPC$^+$) gave rise to LUAD. Lineage tracing analyses demonstrated that although the targeted $Kras^{G12D}$; $Fbxw7^{\Delta/\Delta}$ Club cells showed enhanced proliferation shortly after recombination, activation of squamous markers only occurred after the establishment of SCGB1A1$^+$ hyperplastic lesions[71].

In the LUAD-LUSC transition model driven by expression of KRAS$^{G12D}$ along with STK11 depletion, tumours have been found to originate from SCGB1A1$^+$ Club cells and SFTPC$^+$SCGB1A1$^+$ BASCs, as assessed by allograft assays following 3D organoid culture. Basal cells carrying the same mutations were unable to produce any organoids[69]. It has been suggested that since basal cells express high levels of epidermal growth factor receptor (EGFR), and co-expression of mutant forms of $KRAS$ and $EGFR$ is deleterious[104], activation of oncogenic KRAS in basal cells may lead to oncogene-induced senescence, potentially explaining their inability to propagate either in vitro or in vivo[69].

The GEMM driven by co-ablation of $Pten$ and $Cdkn2ab$, along with $Sox2$ overexpression, combines molecular alterations occurring early during LUSC evolution and is among the few murine models that have been reported to result solely in the formation of squamous lung tumours. Remarkably, these three hits could lead to LUSC development when targeted to KRT14$^+$ basal cells, SCGB1A1$^+$ Club cells, or SFTPC$^+$ Type II cells and/ or BASCs. While tumours mainly arose in the bronchi and bronchioles after targeting KRT14$^+$ cells, they were restricted to peripheral locations when SCGB1A1$^+$ or SFTPC$^+$ cells were targeted. In peripheral pre-invasive lesions, replacement of SCGB1A1 expression by KRT5 occurred as lesions progressed to LUSC, with the two markers showing mutually exclusive patterns[79].

Transplantation studies have shown that following loss of $Keap1$ and $Tp53$ in airway epithelial cells, tumours with features of LUSC only arise from Integrin Alpha 6 (ITGA6)$^+$ basal cells. In contrast to other models where mutant Club cells could give rise to LUSC, tracheal luminal cells carrying the same alterations did not generate any tumours, while peripheral luminal cells gave rise to LUAD[31]. This suggests differential susceptibility among

Table 3 Pre-clinical immunotherapy studies of LUSC.

| Pre-clinical model | Therapy (reference) | Model background | Immune system effect | Tumour effect |
|---|---|---|---|---|
| Syngeneic | Anti-PD1 with WEE1 inhibition [35] | Syngeneic subcutaneous grafts of serially passaged organoid-derived tumours or KLN205 cells in C57BL/6 or DBA/2J mice, respectively. | Reduced accumulation of tumour infiltrating neutrophils. Cytotoxic T cell-mediated tumour cell clearance. | Tumour growth reduction with combined treatment. |
| | Additional conclusions: WEE1 inhibition induces DNA damage that primes endogenous type I interferon and antigen presentation system in LUSC cells. | | | |
| | Anti-PD1, anti-PDL1, and anti-CD137 independently and in combination [83] | Syngeneic subcutaneous grafts of NTCU-induced tumour-derived cell line (UN-SCC680AJ) into A/J mice. | Higher levels of CD45$^+$, CD8$^+$, CD4$^+$, NK, and NKT-cells in tumour cell suspensions were associated with greater tumour response with early dosing of anti-CD137 and anti-PD1 combination treatment. | Anti-PDL1 partially curtailed tumour growth when given within 2 weeks of subcutaneous inoculation; anti-PD1 and anti-CD137 treatment resulted in near-complete tumour rejections. Anti-CD137 and anti-PD1 independently were unable to elicit tumour responses when treatment was delayed >3 weeks after inoculation, yet some tumours responded to combination treatment under the delayed dosing regimen. |
| | Additional conclusions: Immunotherapeutic efficacy of anti-CD137 and anti-PD1 is dependent on the timing of treatment. Delayed treatment resulted in reduced efficacy. CD8$^+$ T cell depletion abrogates the immunotherapeutic ability of the combination treatment of anti-CD137 and anti-PD1 however, CD4$^+$ cell depletion had no effect on treatment efficacy. NK depletion had a small effect on efficacy, with some tumour response abolished. | | | |
| | Chemotherapy and anti-PD1 [85] | KLN205 cells subcutaneously grafted into DBA/2J mice. | LD chemotherapy enhanced CD45$^+$CD3$^+$ and CD45$^+$CD3$^+$CD8$^+$ cytotoxic T cell tumour infiltration. MTD chemotherapy increased immunosuppressive CD11b$^+$F4/80$^+$CD206$^+$ TAMs. | LD chemotherapy increased tumour immunogenicity. Sequential upfront LD chemotherapy and anti-PD1 resulted in greater anti-tumour response than MTD chemotherapy and anti-PD1. |
| | Additional conclusions: LD chemotherapy enhanced tumour antigen exposure partially through the PI3K and NF$\kappa$B pathways. | | | |
| Humanised PDXs | Anti-PDL1 (atezolizumab) Anti-PD1 (pembrolizumab)[124] | Subcutaneous PDXs into humanised mice generated via PBMC or HSPC engraftment. | No significant change in the percentage of hCD45$^+$ hCD3$^+$ cells infiltrating in PDX tumours or in peripheral blood. | Three anti-PDL1 antibodies: atezolizumab, atezolizumab with mutation N298A, and pH-dependent MSB2311 N298A antibody treatment, all resulted in lower tumour volume in a LUSC PBMC-PDX. |
| | Additional conclusions: PBMC-humanised NSG mice better reconstituted the human immune system than HSPC-humanised NSG mice in shorter time, 4-weeks, and 10—14 weeks respectively. PBMC humanised NSG mice facilitated the evaluation of PDL1/PD1 immunotherapy efficacy in patients. | | | |
| Transgenic mice | Anti-PD1 and SX-682 (CXCR1/2 inhibitor) [120] | $Pten^{f/f}$; $Stk11^{f/f}$ mice treated with intratracheal Ad-Cre | Dual treatment increased CD8$^+$ T cells and decreased Ly6G$^+$ neutrophils in the tumour mass. | Combination treatment significantly reduced tumour burden. |
| | Additional conclusions: Ratio of CD8$^+$ T cells and neutrophils predicts immune checkpoint inhibitor efficacy in patients. Induction of IFN-gamma responsive genes (e.g., CXCL10) and relocalisation of lymphocytes. | | | |

*Abbreviations. LD low dose, HSPC human hematopoietic stem and progenitor cell, MTD maximum tolerated dose, NSG NOD/scid/Il2rg$^{-/-}$, PBMC peripheral blood mononucleated cells, TAM tumour associated macrophages.*

distinct subpopulations of airway epithelial progenitor cells to specific genetic hits.

**Tumour microenvironment.** Reciprocal interactions between lung cancer cells and both the stromal and immune components of their microenvironment influence not only the evolution of the tumour[105,106], but also its response to immune-based therapies[7].

PDX-derived cell lines have been applied to 3D co-culture systems with cancer-associated fibroblasts (CAFs) to model interactions with the tumour microenvironment[101]. It was found that forced-expression of SOX2 stimulates transition from hyperplasia to dysplasia and CAFs promote the acquisition of an invasive phenotype, while unexpectedly, suppressing the SOX2-induced dysplastic phenotype.

The angiogenic landscape of LUSC displays intratumour and interpatient heterogeneity. In-depth multi-omic phenotyping of tumour endothelial cells has highlighted the possibility of targeting endothelial subtypes as a potential therapeutic target[107]. In H520 CDXs, the vascular disrupting agent CKD-516 showed enhanced anti-tumour efficacy when administered in combination with radiotherapy compared to single treatments[108].

*Tumour immune microenvironment.* In comparison with LUAD, human LUSC tumours have been found to be enriched in neutrophils, diverse subsets of CD4$^+$ cells—including immunosuppressive regulatory T (Treg) cells—and CD8$^+$ cells expressing programmed cell death (PD1, encoded by *PDCD1*), while containing lower numbers of macrophages[109]. Analyses of the tumour immune microenvironment (TIME) in different GEMMs of LUSC have revealed, in most instances, several similarities with their human counterparts (Table 1).

The presence of large neutrophilic infiltrates has been linked to increased levels of neutrophil recruitment-associated molecules, including CXC ligands, in distinct LUSC models[70,73,78]. Chromatin immunoprecipitation-RNA sequencing analyses on a set of murine LUSC and LUAD tumours, suggested that SOX2 and NKX2-1 exert antagonistic effects on the regulation of *Cxcl5* expression. Indeed, either overexpression of *Sox2* or ablation of *Nkx2-1* in a *Kras*$^{G12D}$; *Tp53*$^{\Delta/\Delta}$ LUAD model—which normally does not contain neutrophilic infiltrates—resulted in both CXCL5 expression and neutrophil recruitment. The ability of SOX2 to induce expression of *CXCL6*, the human orthologue of *Cxcl5*, was conserved in human lung cancer cells in vitro. However, altering the levels of NKX2-1 had no effect on *CXCL6* expression in this setting[78].

In the LUAD-to-LUSC transdifferentiation model driven by *Sox2* overexpression alongside co-deletion of *Stk11* and *Nkx2-1*, it was found that, in contrast to peripheral blood neutrophils, tumour-associated neutrophils (TANs) produce increased levels of ROS and include a subpopulation of cells expressing high levels of sialic acid-binding immunoglobulin-like lectin F (SiglecF)[67]. Both features are suggestive of a tumour-supportive role[78,110,111]. It would be important to determine whether TANs present in tumours from LUSC patients and in murine models exhibiting squamous features throughout LUSC development display similar properties.

High infiltration of CD11b$^+$ myeloid populations has been detected in orthotopic syngeneic LUSC models[35]. The *Chuk*$^{K44A/K44A}$; Tg(*Loricrin-CHUK*) GEMM exhibits extensive infiltration of F4/80$^+$ macrophages within tumours. Macrophage depletion or haematopoietic reconstitution with wild-type bone marrow following irradiation of mutant mice both prevented LUSC development. Macrophage depletion was associated with decreased oxidative stress-induced DNA

damage, suggesting that *Chuk*-mutant macrophages promote tumour development through enhanced ROS production[72].

Immune suppression appears to play a role in murine LUSC progression. Increased levels of several immunosuppressive molecules, including programmed cell death ligand 1 (PDL1, also known as CD274), have been observed in various transgenic LUSC models[71,73,78,79]. Analyses across three different mouse strains demonstrated expression of PDL1 in two-thirds of NTCU-induced LUSC cases and an overall frequency of less than 5% of CD4$^+$ and CD8$^+$ tumour infiltrating lymphocytes; while lymphoid aggregates were evident close to the main bronchi and blood vessels[53]. Additionally, forkhead box P3 (FOXP3)$^+$ Treg cells have been found to be enriched in some murine LUSC tumours[73,78]. In the *Pten*$^{\Delta/\Delta}$; *Stk11*$^{\Delta/\Delta}$ mouse LUSC model, the ratio of CD8$^+$ T cells to Tregs showed an inverse correlation with tumour burden[73]. These attributes emphasise the potential of these autochthonous LUSC models for the evaluation of immune-based therapies.

Bioinformatic studies have identified a subset of human LUSC tumours enriched in inflammatory monocytes (IMs) expressing CD14[81]. In an immune-competent orthotopic murine LUSC model, it was shown that in response to tumour necrosis factor alpha (TNAα)/NFκB pathway activation, LUSC cells secrete C−C motif chemokine ligand 2 (CCL2), leading to IM recruitment. IMs in turn, were found to express high levels of factor XIIIA (FXIIIA, encoded by *F13a1*) and thereby to promote fibrin cross-linking, which favoured invadopodia formation in LUSC cells and metastasis. In human LUSC, increased *F13A1* expression and high levels of intra-tumoural fibrin cross-linking were associated with worse overall and recurrence-free patient survival, respectively[81].

*Application of LUSC models for immunotherapy studies.* Tumour establishment and growth are reliant on the ability to evade immune-mediated clearance. One such way immune evasion can be achieved is tumour-expression of PDL1. PD1 is expressed on T cells and binds to its ligand PDL1 on tumour cells and tumour-associated stromal cells. This results in inhibition of T cell activation and T cell exhaustion, preventing immune-mediated clearance. PD1/PDL1 monoclonal antibodies used effectively in the treatment of NSCLC include: nivolumab, pembrolizumab, atezolizumab, and durvalumab[112]. However, these achieve a response in only ~15−25% of patients with high incidence of immune-mediated side-effects[113]. A better understanding of the mechanisms through which LUSC cells evade immune response is key for the identification of biomarkers for patient stratification and the development of combination therapies that can improve response to current treatments.

Tumours with high mutational load have shown improved responses to immune checkpoint inhibitors in a variety of cancers, including NSCLC[114]. High non-synonymous mutational tumour burden in NSCLC has been associated with better patient response to pembrolizumab[115]. Similarly, the mutational burden of circulating tumour cells in blood has been identified as a potential biomarker for immunotherapy in NSCLC[116]. These observations emphasise the importance for preclinical immunotherapy models to represent a high mutational/neo-antigen load, characteristic of LUSC. LUAD GEMMs have shown a lower non-synonymous mutational load than human adenocarcinomas (0.7 mutations per Mb compared to 1.97 mutations per Mb)[117], highlighting potential limitations of GEMMs for immunotherapy studies.

Both the presence and localisation of T cell populations affect immunotherapy treatment efficacy and patient outcomes. Increased tumour infiltration of FOXP3$^+$ Tregs is associated with a decrease in recurrence free survival in NSCLC patients[118].

Whereas high diversity of PD1+CD8+ T cell repertoires in peripheral blood has been associated with improved response to anti-PD1/PDL1 therapies and increased progression-free survival[119].

High levels of neutrophil infiltration predict poor response to immunotherapy in a subset of myeloid-rich NSCLC tumours. Using multiplexed-IHC, it was found that the ratio of CD8+ T cells to neutrophils within the tumour mass was able to distinguish between patients who were responsive to immune checkpoint inhibitors and those with stable or progressive disease[120]. In the *Pten*- and *Stk11*-deficient GEMM LUSC model, inhibition of neutrophil's CXC chemokine receptors 1 and 2 (CXCR1/2)—which regulate their recruitment to the TIME[110]—combined with anti-PD1 treatment, decreased tumour masses, unlike single therapy treatment[120]. This was associated with a reduction of Lymphocyte antigen 6 complex locus G (Ly6G)+ neutrophils in the tumour mass and increased relocation of CD8+ T cells from the tumour periphery into the tumour mass, suggesting that TANs impede CD8+ T cell infiltration[120].

The combination of targeted inhibitors alongside anti-PD1 treatment has shown increased tumour responses, thought to be mediated by increased immune system activation. Inhibition of WEE1 G2 checkpoint kinase (WEE1), a CDK1 negative-regulator, alongside PD1 blockade resulted in diminished tumour growth in two LUSC subcutaneous syngeneic models. The associated increase in DNA damage triggered by WEE1 inhibition and consequent CDK1 activation stimulated expression of type 1 interferon (IFN) signalling and major histocompatibility complex (MHC) class I antigen presentation genes in LUSC cells, increased tumour-infiltrating CD8+ T cells, and decreased TANs[35]. This demonstrates the enhanced efficacy of combination therapies in the treatment of LUSC by further unleashing the immune system. In renal, colon, and prostate syngeneic cancer models, orthotopic implanted grafts had a lesser response to immunotherapeutic agents when compared to subcutaneous implants[121]. It would be important to address whether syngeneic LUSC models display a similar behaviour, so that the implantation site can be considered during experimental design.

The creation of humanised PDXs allows modelling of human immune and stromal interactions in mice[122], enabling the application of xenograft models to lung immunotherapy studies. Humanised murine models have been generated by reconstituting the immune system of mice with either human peripheral blood mononucleated cells (PBMCs) or haematopoietic stem and progenitor cells (HSPCs). PBMC implantation has been shown to reconstitute the immune system faster than HSPC implantation (4 weeks as opposed to 10−14 weeks). Studies utilising humanised NSCLC CDX/PDX models have demonstrated the ability of anti-PD1 and anti-PDL1 to slow tumour growth in vivo[123,124] (Table 3), presenting new opportunities for combination trials.

**Conclusions and perspectives**. Molecular profiling analysis of patient samples corresponding to different developmental stages of LUSC has provided insights into the alterations underlying cancer initiation and progression[5,6,14,105,106]. This has led to the identification of candidate driver genes and biomarkers, and fuelled the refinement of LUSC models to more closely mimic human pathology. Advances in cell culture methods for airway epithelial cells, such as 3D organoid cultures, and in genetic manipulation technologies, including CRISPR/Cas9 gene editing, have further contributed to streamlining lung cancer modelling. An increasingly growing array of in vitro and in vivo systems, each with their own strengths and limitations, are now used for both fundamental and translational LUSC research.

A major area of the current investigation is the interactions between LUSC cells and their microenvironment, particularly the TIME[7]. Due to their immune proficiency, the great majority of studies have thus far utilised GEMMs and murine syngeneic models. Yet, the development of humanised PDXs and co-culture systems is opening new avenues for research using human material. For example, autologous co-cultures of tumour-derived organoids and PBMCs from NSCLC patients, including one case of LUSC, have been used to assess the ability of T cells to recognise and eliminate tumour cells[125]. In LUAD, the high relative expression of receptor tyrosine kinase-like orphan receptor-1 (ROR1) in lung tumours to normal tissue has been exploited by engineering ROR1-specific chimeric antigen receptor (CAR) T cells, showing promising results in both static and dynamic co-culture systems with A549 LUAD cells[126].

With programmes for early cancer detection and intervention becoming top priorities, the availability of models recapitulating the pre-invasive stages of LUSC is essential for the identification of relevant biomarkers and therapeutic targets. NTCU-induced murine LUSC currently is one of the best characterised models in this sense. A better understanding of the genetic events underlying NTCU-induced carcinogenesis will be key for its optimal use in preclinical studies. While the combination of genetic manipulation techniques, pharmacological modulators, and 2D and 3D co-culture systems is expected to support the creation of novel human premalignant LUSC in vitro models.

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

## Acknowledgements

We want to thank Marie-Belle El Mdawar, Lukas Kalinke, Kyren Lazarus (all from UCL Respiratory), and Rob Hynds (UCL Cancer Institute) for helpful comments during the preparation of this manuscript, and Rebecca Graham (UCL Respiratory) for proof-reading. This work was funded by the Wellcome Trust [WT107963AIA] and a CRUK programme award to S.M.J. S.M.J. is further supported by the Roy Castle Lung Cancer Foundation, the Rosetrees Trust, UCLH Charitable Foundation, the Longfonds BREATH consortium, and the MRC UKRMP2 cell therapy platform. Z.E.W. is funded by a CRUK-UCL Centre PhD studentship [A27437].

## Author contributions

S.G.-L. and Z.E.W wrote the paper with input from S.M.J.

## Competing interests

Paid advisory boards: 2018 and 2020 Astra-Zeneca; 2018 and 2019 BARD1 Life Sciences; 2018 Achilles Therapeutics; 2019 Johnson and Johnson (S.M.J.). Assistance or travel to meetings: Astra-Zeneca, ATS 2018 San Diego; Takeda, WCLC 2019 Barcelona (S.M.J.). Grant Investigator lead 2018-2028 GRAIL Inc; 2018 Owlstone (S.M.J.). Personal: Spouse works for and owns shares in Astra-Zeneca (S.M.J.). S.G.-L. and Z.E.W. declare no competing interests relating to this work.
