## [Peer Review File · Communications Biology]

Reviewers' comments:

Reviewer #1 (Remarks to the Author):

General: In the manuscript by Gómez-López et al entitled "Mapping lung squamous cell carcinoma pathogenesis through in vitro and in vivo models" provides a much-needed overview in the field of LUSC models. The review is well written and does a good job balancing the current, or past, model systems used to study LUSC. I'm pleased someone set out to cover this important topic in a largely ignored but very important lung cancer subtype. This will have a large interest in the field and the authors should be applauded for the thorough manuscript. I have some specific comments that I hope will help this be even more complete and serve as an excellent review that is fully up-to-date.

Specific comments:

- There are several sections I'd suggest adding to the citations to be more complete given several important contributions in the field:

o Cell line generation: Adi Gazdar and John Minna generated the vast majority of NSCLC lines used throughout the world. They wrote a helpful paper "Gazdar et al, JNCI, 2010" that discusses some of the challenges they faced at NCI with generating LUSC lines. I suggest adding this citation and comment on their findings, which may inform others trying to establish such lines.

o GEMM section: Although cited and listed in the table, I believe the paper by "Zuoxiang Xiao et al, Cancer Cell, 2013" certainly deserves discussion amongst the other GEMM already well detailed. This paper had 100% incidence of LUSC, unlike many of the other GEMMs already covered. Of note another murine cell line, KAL, was also developed in this paper and it's mention should be added to the review since it grows in immune-competent mice (FVB). Additionally, I think it should be commented on that several of the GEMMs featured in this review that utilize Stk11/Lkb1 deletion, which yield mixed histology lung cancers, may not faithfully recapitulate LUSC because Stk11/Lkb1 mutations are actually quite rare in LUSC. These are much more common in lung adenocarcinomas. The notable immune TME changes from Stk11 mutations that occur in these GEMMs appear related to the neutrophil populations, which again may not recapitulate in human LUSCs.

o One section worth considering either adding or incorporating into other in vivo sections is the study of metastases. Few of the mentioned models form distant metastases, but in vivo selection techniques have led to more highly metastatic subclones of the already mentioned parental KLN205 and KAL lines (Porrello et al, Nature Communications, 2018). Recently developed metastatic human LUSC cell lines SKMES-LN1 and H520-LN3 (Harrison et al, Cancer Research, 2020) have demonstrated spread to distant extra-thoracic sites common to clinical disease (e.g. adrenal glands). Additionally the deletion of Nkx2-1 in the SOX2/Lkb1 GEMM (Mollaoglu et al, Immunity, 2018) substantially decreased latency and appeared to increase metastatic potential.

o TME Section: There is no mention of tumor vasculature. Some discussion of studies (or lack of studies) on angiogenesis is warranted, given that angiogenesis inhibitors have a clinical history in LUSC.

o Tumor Immune Microenvironment Section: The above model mentioned by Xiao et al should also be added to this section as they found it was characteristic of tumor associated macrophages. Dense infiltration of myeloid cells were also found in the organoid-transplantation model described by Han et al, Clinical Cancer Research, 2020 (citation #33).

o Immunotherapy Section: Notably because LUSC is a disease of smokers, it is important to emphasize that use of GEMMs to evaluate immunotherapies is problematic as these models do not have nearly the neo-antigen/mutational load as real human disease. This is well demonstrated by the paper by McFadden et al, PNAS, 2016. We suggest adding this citation and adding this limitation, which many have acknowledged in the field.

o For Fig 2 it would be helpful to add a panel depicting the cell signaling pathways altered by the listed oncogenes and tumor suppressors. It is mentioned in the text that many of these are linked (for example, that SOX2 and ECT are targets of PRKCI, p5) and a schematic showing the relationships between genetic alterations in LUSC would help the reader follow the text and serve as a useful reference.

Reviewer #2 (Remarks to the Author):

Comms Biology Review of COMMSBIO-21-0288

The review submitted by Dr. Janes and colleagues provides a comprehensive examination of in vitro and in vivo models used to study lung squamous cell carcinoma (LUSC). Overall the authors did a very nice job presenting a well-written and thorough review examining models of LUSC. They covered most relevant models available of human and mouse LUSC, both in vitro and in vivo. The authors did spend too much time on targeted therapy, but this acceptable given the focus of the review is more on genetic models, cells of origin, and interactions between tumor cells and immune cells. The authors did a thorough job considering landscape of squamous cell biology and its molecular regulation. The figures are straightforward and easy to follow. The cell culture models in Figure 1 should be annotated better to differentiate the cell culture models. The font in Figure 2 needs to be increased as it is hard to read. The references look mostly on point.

Reviewer #3 (Remarks to the Author):

In the manuscript Sam Janes and colleagues have reviewed lung squamous cell carcinoma (LUSC). This is an excellent review that thoroughly covers LUSC model systems, their known pros and cons, the biological processes that promote LUSC, and cell of origin. There are no papers or topics that this reviewer is aware of that have been omitted, and citations support the points being made by the authors.

Minor comments

Line 551: "This was associated a reduction of Lymphocyte..." should read "This was associated with a reduction of Lymphocyte..."

We want to thank the reviewers for their encouraging feedback and suggestions. They were thorough and helpful and we believe have improved this work.

In addition to addressing the comments from the reviewers, we have added new references from works that were published around the date that the first version of this manuscript was submitted (Torres-Ayuso *et al. Cancer Discovery*, 2021; Yuan *et al. Nature*, 2021; Woo *et al. Nat Genet*, 2021). We also edited the text to ensure it was within the 7000 words limit. All changes and additions are indicated in red in the manuscript.

All abbreviations used in the tables are defined in the legends included below the relevant table.

Below we provide a point-by-point response to each of the reviewers' comments.

C: Comment

R: Response

Reviewers' comments:

Reviewer #1 (Remarks to the Author):

General: In the manuscript by Gómez-López et al entitled "Mapping lung squamous cell carcinoma pathogenesis through in vitro and in vivo models" provides a much-needed overview in the field of LUSC models. The review is well written and does a good job balancing the current, or past, model systems used to study LUSC. I'm pleased someone set out to cover this important topic in a largely ignored but very important lung cancer subtype. This will have a large interest in the field and the authors should be applauded for the thorough manuscript. I have some specific comments that I hope will help this be even more complete and serve as an excellent review that is fully up-to-date.

Specific comments:

- There are several sections I'd suggest adding to the citations to be more complete given several important contributions in the field:

C1. Cell line generation: Adi Gazdar and John Minna generated the vast majority of NSCLC lines used throughout the world. They wrote a helpful paper "Gazdar et al, JNCI, 2010" that discusses some of the challenges they faced at NCI with generating LUSC lines. I suggest

adding this citation and comment on their findings, which may inform others trying to establish such lines.

R1. We agree with the reviewer that this work by Adi Gazdar and colleagues provides useful insights and advice that should be taken forward when using NSCLC cell lines. The reference (#3) has been added in lines 27-30:

'However, in the absence of molecular fingerprinting and routine surveillance, intra- or inter-species contamination and cell line misidentification can severely compromise the potential of these models³.'

C2. GEMM section: Although cited and listed in the table, I believe the paper by "Zuoxiang Xiao et al, Cancer Cell, 2013" certainly deserves discussion amongst the other GEMM already well detailed. This paper had 100% incidence of LUSC, unlike many of the other GEMMs already covered. Of note another murine cell line, KAL, was also developed in this paper and it's mention should be added to the review since it grows in immune-competent mice (FVB). Additionally, I think it should be commented on that several of the GEMMs featured in this review that utilize *Stk11/Lkb1* deletion, which yield mixed histology lung cancers, may not faithfully recapitulate LUSC because *Stk11/Lkb1* mutations are actually quite rare in LUSC. These are much more common in lung adenocarcinomas. The notable immune TME changes from *Stk11* mutations that occur in these GEMMs appear related to the neutrophil populations, which again may not recapitulate in human LUSCs.

R2. These are important points. We have now covered the model described by Xiao *et al.* in the main text under the GEMM section (lines 293-300): 'Although rarely affected by genetic alterations, the serine/threonine kinase component of inhibitor of nuclear factor kappa B kinase complex (*CHUK*, also known as *IKKA*) is downregulated in a fraction of human LUSC cases. Knock-in mice expressing a kinase-dead form of *CHUK* (*Chuk^{K44A}*) develop severe skin lesions and spontaneous LUSC⁷². Re-introduction of wild-type *Chuk* under the control of the *Loricrin* promoter rescued the skin phenotype, while LUSC developed with 100% penetrance. Tumours exhibited elevated ERK activity and decreased p53 levels and their formation was preceded by increased expression of Δ Np63 and tripartite motif containing 29 (TRIM29), inflammation and lung enlargement.'

The KAL^{LU} cell line and its metastatic derivative KAL-LN2E1 have been included in Table 2. We have also added the name of one of the lines already included and indicated the strain background of all the syngeneic models listed in the table.

We agree with the reviewer that the GEMMs that include *Stk11* loss-of-function may not be fully representative of human LUSC. Indeed, throughout the text we highlight that in many of these models LUSC arises through differentiation of lesions originally with features of LUAD

(lines 272-274; 323-325; 454; 515-516). For clarity, we have now added the following sentence after the first mention of the use *Stk11* inactivation in a LUSC GEMM (lines 277-278): 'However, mutations or downregulation of *STK11* are rare in human LUSC (Fig. 3) and more prevalent in LUAD³⁸.'

Regarding the features and role of tumour infiltrating neutrophils in the LUAD-LUSC transdifferentiation model induced by co-deletion of *Stk11* and *Nkx2-1*, we agree with the reviewer that it may not be conserved in human LUSC, and had indicated this in the text (lines 519-521): 'It would be important to determine whether TANs present in tumours from LUSC patients and in murine models exhibiting squamous features throughout LUSC development display similar properties'. The enrichment of neutrophils in human LUSC tumours, however, has been reported in the work by Kargl *et al. Nat Commun*, 2017, which we reference in lines: 498-501 (Reference #109).

C3. One section worth considering either adding or incorporating into other in vivo sections is the study of metastases. Few of the mentioned models form distant metastases, but in vivo selection techniques have led to more highly metastatic subclones of the already mentioned parental KLN205 and KAL lines (Porrello *et al*, *Nature Communications*, 2018). Recently developed metastatic human LUSC cell lines SKMES-LN1 and H520-LN3 (Harrison *et al*, *Cancer Research*, 2020) have demonstrated spread to distant extra-thoracic sites common to clinical disease (e.g. adrenal glands). Additionally the deletion of *Nkx2-1* in the *SOX2/Lkb1* GEMM (Mollaoglu *et al*, *Immunity*, 2018) substantially decreased latency and appeared to increase metastatic potential.

R3. We appreciate that although we included information about metastasis in the GEMMs listed in Table 1, in the main text we focused on primary tumour models. We have now incorporated the works suggested by the reviewer into different sections of the manuscript to be able to keep the same structure.

The work by Porrello *et al. Nat Commun*, 2018 (Reference #81) has now been included in Table 2 and cited in the main text of the syngeneic models section (line 353) and described in the TIME section (lines 541-548): 'Bioinformatic studies have identified a subset of human LUSC tumours enriched in inflammatory monocytes (IMs) expressing CD14⁸¹. In an immune-competent orthotopic murine LUSC model, it was shown that in response to tumour necrosis factor alpha (TNA α)/NF κ B pathway activation, LUSC cells secrete C-C motif chemokine ligand 2 (CCL2), leading to IM recruitment. IMs in turn, were found to express high levels of factor XIII A (FXIII A, encoded by *F13a1*) and thereby to promote fibrin cross-linking, which favoured invadopodia formation in LUSC cells and metastasis. In human LUSC, increased *F13A1*

expression and high levels of intra-tumoural fibrin cross-linking were associated with worse overall and recurrence-free patient survival, respectively⁸¹.'

We have referenced the publication by Harrison *et al. Cancer Res*, 2020 (Reference #89) in the xenografts section (lines 368-371): 'Serial passaging of orthotopic CDXs derived from the LUSC cell lines H520 and SKMES-1 has produced subclones that metastasise to LUSC characteristic sites (*i.e.* adrenal glands and chest wall)⁸⁹ and could be used as a xenograft models of LUSC metastasis.'

Regarding the work by Mollaoglu *et al. Immunity* 2018, we had indicated in the text that 'co-deletion of *Nkx2-1* in this model [driven by *Sox2* overexpression of coupled with loss of *Stk11*], significantly reduced tumour latency' (line 322). However, we were unable to find any information about metastasis in this model in that publication.

C4. TME Section: There is no mention of tumor vasculature. Some discussion of studies (or lack of studies) on angiogenesis is warranted, given that angiogenesis inhibitors have a clinical history in LUSC.

R4. We have now discussed this topic (lines 491-495), including a new reference (#107: Goveia *et al. Cancer Cell*, 2020) and a work we had already referred to (#108): 'The angiogenic landscape of LUSC displays intratumour and interpatient heterogeneity. In depth multi-omic phenotyping of tumour endothelial cells has highlighted the possibility of targeting endothelial subtypes as a potential therapeutic target¹⁰⁷. In H520 CDXs, the vascular disrupting agent CKD-516 showed enhanced anti-tumour efficacy when administered in combination with radiotherapy compared to single treatments¹⁰⁸.'

C5. Tumor Immune Microenvironment Section: The above model mentioned by Xiao et al should also be added to this section as they found it was characteristic of tumor associated macrophages. Dense infiltration of myeloid cells were also found in the organoid-transplantation model described by Han et al, *Clinical Cancer Research*, 2020 (citation #33).

R5. We have followed the reviewer's advice and covered these two works within the TIME section (lines 523-528): High infiltration of CD11b⁺ myeloid populations has been detected in orthotopic syngeneic LUSC models³⁵. The *Chuk*^{K44A/K44A};Tg(*Loricrin-CHUK*) GEMM exhibits extensive infiltration of F4/80⁺ macrophages within tumours. Macrophage depletion or haematopoietic reconstitution with wild-type bone marrow following irradiation of mutant mice both prevented LUSC development. Macrophage depletion was associated with decreased oxidative stress-induced DNA damage, suggesting that *Chuk*-mutant macrophages promote tumour development through enhanced ROS production⁷².

C6. Immunotherapy Section: Notably because LUSC is a disease of smokers, it is important to emphasize that use of GEMMs to evaluate immunotherapies is problematic as these models do not have nearly the neo-antigen/mutational load as real human disease. This is well demonstrated by the paper by McFadden et al, PNAS, 2016. We suggest adding this citation and adding this limitation, which many have acknowledged in the field.

R6. We thank the reviewer for highlighting this very important point. We have incorporated the work by McFadden *et al. PNAS*, 2016 (Reference #117) within the second paragraph of the immunotherapy section, which now reads (lines 562-570): 'Tumours with high mutational load have shown improved responses to immune checkpoint inhibitors in a variety of cancers, including NSCLC¹³. High non-synonymous mutational tumour burden in NSCLC has been associated with better patient response to pembrolizumab¹⁴. Similarly, the mutational burden of circulating tumour cells in blood has been identified as a potential biomarker for immunotherapy in NSCLC¹⁵. These observations emphasize the importance for preclinical immunotherapy models to represent a high mutational/neo-antigen load, characteristic of LUSC. LUAD GEMMs have shown a lower non-synonymous mutational load than human adenocarcinomas (0.7 mutations per Mb compared to 1.97 mutations per Mb)¹⁶, highlighting potential limitations of GEMMs for immunotherapy studies.'

C7. For Fig 2 it would be helpful to add a panel depicting the cell signaling pathways altered by the listed oncogenes and tumor suppressors. It is mentioned in the text that many of these are linked (for example, that SOX2 and ECT are targets of PRKCI, p5) and a schematic showing the relationships between genetic alterations in LUSC would help the reader follow the text and serve as a useful reference.

R7. We thank the reviewer for highlighting how useful the addition of a schematic displaying the signalling interactions discussed would be to readers. We have added the following figure to the review (now Figure 2) with additional supporting references included in the figure legend.

Figure 2. Interaction of signalling pathways demonstrated in *in vivo* and *in vitro* models of LUSC. SOX2, ECT2, PKC ι (encoded by *PRKCI*) and PI3K signalling cooperate to promote a neoplastic cell fate in LUSC models. AKT is a downstream effector of stimulated p110 α . Full AKT activation is achieved when phosphorylated at both positions S473 and T308¹²⁷. High levels of SOX2 have been correlated with upregulated phospho-AKT²¹. PKC ι phosphorylates and directly interacts with ECT2 to promote anchorage independent growth invasion through downstream targets. PKC ι phosphorylates SOX2 favouring squamous cell fate and decreased differentiation¹⁶. Loss of p53, PTEN and KEAP1 have been used to model LUSC phenotypes both *in vitro* and *in vivo*. Simultaneous loss of p53 and KEAP1 has shown synergistic effects, inducing increased proliferation, metastatic potential and resistance to oxidative stress³¹. p53 activity can inhibit PI3K signalling through PTEN-dependent and potentially -independent mechanisms in squamous cell carcinomas¹²⁸. Additional interactions between depicted proteins have been described in other cellular contexts.

Reviewer #2 (Remarks to the Author):

Comms Biology Review of COMMSBIO-21-0288

C. The review submitted by Dr. Janes and colleagues provides a comprehensive examination of *in vitro* and *in vivo* models used to study lung squamous cell carcinoma (LUSC). Overall the authors did a very nice job presenting a well-written and thorough review examining models of LUSC. They covered most relevant models available of human and mouse LUSC, both *in vitro* and *in vivo*. The authors did spend too much time on targeted therapy, but this acceptable

given the focus of the review is more on genetic models, cells of origin, and interactions between tumor cells and immune cells. The authors did a thorough job considering landscape of squamous cell biology and its molecular regulation. The figures are straightforward and easy to follow. The cell culture models in Figure 1 should be annotated better to differentiate the cell culture models. The font in Figure 2 needs to be increased as it is hard to read. The references look mostly on point.

R. We thank the reviewer for her/his positive and constructive feedback. We have updated Figure 1 and its figure legend to add clarity about the different models included:

Figure 1. *In vitro* models of lung cancer and their application in *in vivo* studies. Establishment of **air-liquid-interface (ALI)** and organoid cultures from human or mouse airway epithelial cells (left) and LUSC tissue (right). **Following ALI or 3D culture, normal airway epithelial basal cells produce pseudostratified epithelial sheets or hollow organoids containing differentiated cells, respectively. In contrast, LUSC cells give rise to epithelial sheets with features of dysplasia and more solid, disorganised organoids.** Cultured cells may be subjected to genetic and pharmacological manipulation to investigate the phenotypic consequences of molecular alterations recurrently identified in LUSC samples. Organoids can be used in *in vitro*

drug screenings and may be implanted into mice to evaluate their ability to give rise to tumours *in vivo* and response to therapies. ECM, extracellular matrix.

We have also created a new version of Figure 2 (now Figure 3). We have increased the size of the font for easier readability and also used 'normal adjacent tissue' as reference cohort rather than 'diploid samples' as we had used in the first version. We believe this new version provides more relevant information to the reader. As a result of this change, we updated the percentages related to the frequency of *SOX2* upregulation in the main text (line 316).

Reviewer #3 (Remarks to the Author):

In the manuscript Sam Janes and colleagues have reviewed lung squamous cell carcinoma (LUSC). This is an excellent review that thoroughly covers LUSC model systems, their known pros and cons, the biological processes that promotes LUSC, and cell of origin. There are no papers or topics that this reviewer is aware of that have been omitted, and citations support the points being made by the authors.

We are grateful to the reviewer for such positive feedback.

Minor comments

C. Line 551: "This was associated a reduction of Lymphocyte..." should read "This was associated with a reduction of Lymphocyte..."

R. We thank the reviewer for pointing this out. We have corrected the text and it now reads: "This was associated **with** a reduction of Lymphocyte antigen..."

REVIEWERS' COMMENTS:

Reviewer #1 (Remarks to the Author):

The authors have done an excellent job in response to my prior comments and I have nothing further to add. It's an excellent review and will be an invaluable resource for the community!
Bravo!